# Swallowing Functions after Sagittal Split Ramus Osteotomy with Loose Fixation for Mandibular Prognathism: A Retrospective Case Series Research

**DOI:** 10.3390/ijerph20031926

**Published:** 2023-01-20

**Authors:** Kei-ichiro Miura, Masashi Yoshida, Satoshi Rokutanda, Takamitsu Koga, Masahiro Umeda

**Affiliations:** 1Department of Clinical Oral Oncology, Nagasaki University Graduate School of Biomedical Sciences, Nagasaki 852-8588, Japan; 2Department of Oral Surgery, Imakiire General Hospital, Kagoshima 890-0051, Japan; 3Department of Dentistry and Oral Surgery, Juko Memorial Nagasaki Hospital, Nagasaki 852-8004, Japan

**Keywords:** orthognathic surgery, sagittal split ramus osteotomy, swallowing function, mandibular prognathism

## Abstract

Sagittal split ramus osteotomy (SSRO) is a standard surgical technique for patients with mandibular prognathism. However, the appropriate position of the proximal fragment is not strictly defined, and rigid fixation can induce early postoperative skeletal relapse and temporomandibular (TMJ) disorders. Loose fixation can be expected to seat the proximal bone fragments in a physiologically appropriate position, thereby reducing adverse events. Although long-term skeletal stability has been achieved using SSRO without fixation, the evaluation of preoperative and postoperative eating and swallowing functions remains unclear, and this study aimed to clarify this point. We evaluated mastication time, oral transfer time, and pharyngeal transfer time using videofluorography (VF) preoperatively, two months postoperatively, and six months postoperatively, and along with the position of anatomical landmarks using cephalometric radiographs, modified water swallowing test (MWST), food test (FT), and repetitive saliva swallowing test (RSST) were used to evaluate postoperative swallowing function. Four patients (one male, three females; mean (range) age 26.5 (18–51) years) were included, with a mean setback of 9.5 mm and 6.5 mm on the right and left sides, respectively. Postoperative eating and swallowing functions were good in VF, cephalometric analysis, MWST, FT, and RSST. In the present study, good results for postoperative eating and swallowing functions were obtained in SSRO with loose fixation of the proximal and distal bone segments.

## 1. Introduction

Sagittal split ramus osteotomy (SSRO) is a common surgical procedure for the treatment of mandibular prognathism, and postoperative skeletal stability has generally been established [1]. However, since the appropriate position of the postoperative proximal segment is not strictly defined, SSRO causes postoperative skeletal relapse and malocclusion when the proximal and distal segments are unfavorably fixed [2]. Previously, condylar positioning devices were used to reposition the proximal segment to the original position [3]. However, Costa et al. reported a lack of evidence for the efficiency of this equipment [4]. Intraoral vertical ramus osteotomy (IVRO) is a major surgical treatment for jaw deformity and has been shown to be effective in leading the proximal segment to a mechanically balanced position [5]. The drawbacks of IVRO include less contact between the proximal and distal segments than SSRO, which requires more time for bone healing, and the proximal fragment is bounced buccally by the distal segment [6].

To overcome these drawbacks of SSRO or IVRO, a novel surgical treatment method for placing the proximal segment in a physiological position without fixation of the proximal and distal segments by plates and screws was suggested, and long-term skeletal stability was obtained [7]. Patients with skeletal mandibular prognathism may have postoperative narrowing of the oropharyngeal region, which may affect eating and swallowing functions [8]. Therefore, it is necessary to examine how eating and swallowing functions change postoperatively. Previously, there has been a study of eating and swallowing function in patients with skeletal mandibular prognathism who underwent SSRO with fixation [9]. However, there has been no evaluation of postoperative eating and swallowing function in SSRO with loose fixation. Regarding the evaluation of eating and swallowing, methods for evaluating dysphagia using salivary, water, and food swallowing have already been established in the head and neck region, as well as methods for evaluating function from the oral cavity to the pharynx using videofluorography [10]. Therefore, these evaluation methods can also be applied to preoperative and postoperative patients with mandibular prognathism.

The objective of this study was to examine postoperative eating and swallowing functions in patients with mandibular prognathism that underwent SSRO with loose fixation.

## 2. Materials and Methods

### 2.1. Patient (Table 1)

We studied four patients (one male, three females; mean (range) age 26.5 (18–51) years) that underwent SSRO with loose fixation for the correction of mandibular prognathism between April 2013 and September 2013. The mean setback was 9.5 mm and 6.5 mm on the right and left sides, respectively. Inclusion criteria were defined as patients with skeletal mandibular prognathism who had completed the growth phase and had not undergone orthognathic surgery in the past. Exclusion criteria were defined as patients who had undergone orthognathic surgery in the past, had a serious systemic disease, or suffered an unexpected intraoperative fracture.

**Table 1 ijerph-20-01926-t001:** Details of patients who took part.

Patient No.	Gender (Male/Female)	Age (Years)	Setback: Right (mm)	Setback: Left (mm)
1	Female	18	12	12
2	Female	18	7	0
3	Male	19	12	9
4	Female	51	7	5

### 2.2. Surgical Technique and Postoperative Management

We performed a modified SSRO (short lingual osteotomy) as previously reported by Hunsuck [11] and Epker [12]. All operations were performed under general anesthesia, and no fixation with screws or plates was performed. After bone split, the bone segments were ligated with 2-0 polyglactin 910 sutures (Vicryl; Ethicon, Somerville, NJ, USA), and intermaxillary fixation with elastics was applied for two weeks. Subsequently, a postoperative orthodontic treatment was initiated with postoperative jaw exercise. The protocol for this study was approved by the Medical Ethics Committee of Imakiire General Hospital (Reference No. 105). This study was performed in accordance with the Declaration of Helsinki and subsequent amendments [13]. All patients were fully informed about the procedures and provided written informed consent prior to enrollment.

### 2.3. Analysis of Cephalometric Photographs

Cephalometric photographs (Figure 1) were taken to assess both skeletal stability and changes in pharyngeal soft tissue morphology, with reference to a previous study [9].

The position of the maxilla and mandible relative to the base of the cranium was defined as (1) SNA: sella-nasion-A point (SNA) angle, and (2) SNB: sella-nasion-B point (SNB) angle. The anteroposterior position of the maxilla and mandible was defined as (3) ANB: A point-nasion-B point (ANB) angle. The pre- and postoperative position of the hyoid bone was defined as (4) HSN: Angle between the line connecting Sella (S) and the lowest point of the hyoid bone (H) and the SN plane, (5) S-H (mm): Distance from S to H, (6) C3-H (mm): distance from the lowest point of the anterior third cervical vertebra to H, and (7) PNS-T (mm): Distance between the PNS and the dorsal surface of the tongue on a line perpendicular to the FH plane and passing through the PNS. The anteroposterior width of the pharyngeal region was defined as (8) P-T (soft palate) (mm): the distance between the tongue and the posterior wall of the pharynx on a line parallel to the FH plane through the tip of the soft palate, and (9) P-T (epiglottis) (mm): the distance between the tongue and the posterior wall of the pharynx on a line parallel to the FH plane through the tip of the soft palate.

**Figure 1 ijerph-20-01926-f001:**
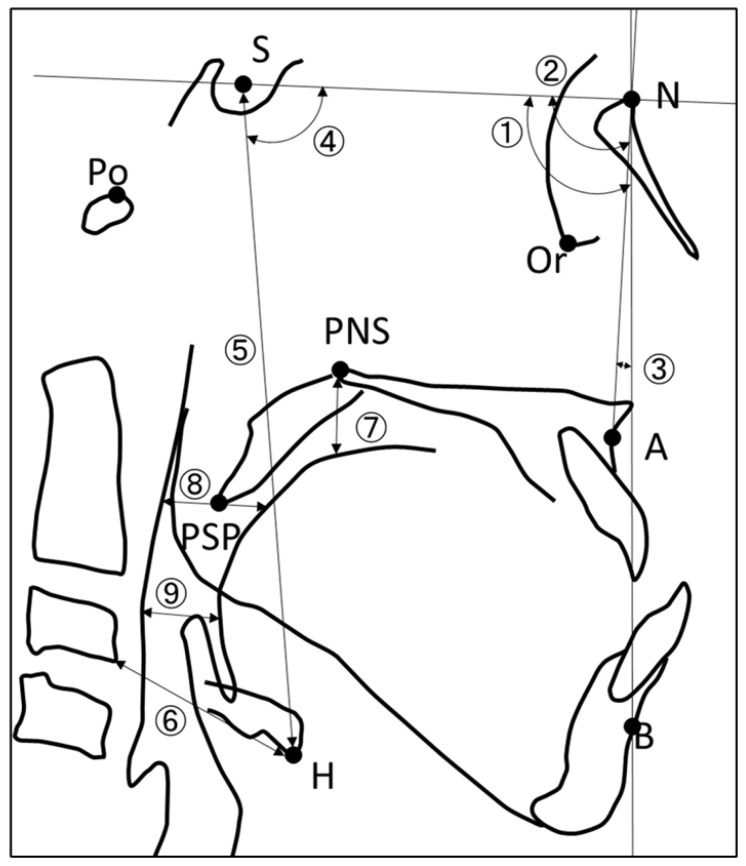
Analysis of cephalometric photographs.

Position of the maxilla and mandible to the base of the cranium

(1). SNA, (2). SNB,

Anteroposterior position of the maxilla and mandible

(3). ANB

Pre- and postoperative position of the hyoid bone and tongue

(4). HSN, (5). S-H, (6). C3-H, (7). PNS-T

Anterior-posterior width of pharyngeal region

(8). P-T (soft palate), (9). P-T (epiglottis)

### 2.4. Videofluorography (VF)

VF was used to assess the swallowing function in all patients. Since a cookie soaked with barium (barium cookie) was used to evaluate the ability of food mastication and swallowing, half of the barium cookie was projected with VF, and mastication, eating, and swallowing were evaluated [14]. VF was assessed by measuring mastication, oral transfer, and pharyngeal transfer times, and the definitions of each data point were based on the following: Mastication time was defined as the time from the start of mastication until the hyoid bone began to elevate. Oral transfer time was defined as the time from the start of hyoid bone elevation until the bolus passed the posterior border of the ramus mandibularis. Pharyngeal transfer time was defined as the time from the moment the bolus passed the posterior border of the ramus mandibularis until it passed through the esophageal inlet. Mastication, oral transfer, and pharyngeal transfer time were each measured four times, and the average value was calculated.

### 2.5. Modified Water Swallowing Test (MWST)

The MWST was performed to assess pharyngeal function and aspiration based on the swallowing motion and profile. MWST was performed as described below. The patient was asked to swallow 3 mL of cold water applied to the oral floor, and the evaluator assessed swallowing. At that time, choking, changes in respiratory status, and wet hoarseness were assessed. If there was no wet hoarseness, two additional dry swallows were performed.

The evaluation criteria were defined as follows

Score 1 (Very poor): No swallowing, choking, and/or respiratory distress

Score 2 (Poor): With swallowing and respiratory distress

Score 3 (Fair): With swallowing, good respiration, choking, and/or wet hoarseness

Score 4 (Good): Swallowing, good respiration, and no choking

Score 5 (Excellent): Grade 4 plus repetitive swallowing within 30 s

A score of 3 or less was defined as a problem with swallowing, and if the score was 4 or more, it was repeated a maximum of two more times, with the worst result being the grade.

### 2.6. Food Test (FT)

FT was performed to evaluate the oral food residue after swallowing. One spoonful (4 g) of the pudding was placed on the tongue and assessed according to the diagnostic criteria based on the MWST, with the addition of the following items: presence of oral residues (Score 3) and almost no oral residues (Score 4).

### 2.7. Repetitive Saliva Swallowing Test (RSST)

The RSST is a simple and safe assessment of swallowing function that evaluates saliva swallowing in 30 s. The procedure is as follows: (1) the examiner places a finger on the patient’s pharyngeal prominence and hyoid, (2) patients are instructed to repeat the salivary swallow repeatedly for 30 s, and (3) the elevation of the pharyngeal prominence is counted when the patients swallow saliva. If the number of swallows was less than three, a decline in swallowing function was suspected.

### 2.8. Timing of Assessment

Radiological and clinical assessments were performed immediately before surgery (T1), two months (T2), and six months (T3) after surgery.

### 2.9. Data Analysis Methods

Analysis of cephalometric photographs was performed using OsiriX MD software (version 12.0.3; Pixmeo SARL, Geneva, Switzerland), and VF was evaluated using PowerDVD (version 12.0.6708.55; CyberLink Corp, Taipei, Taiwan). Both analyses were performed by a skilled oral and maxillofacial surgeon with at least 10 years of experience.

## 3. Results

### 3.1. Analysis of Cephalometric Photographs

As for the position of both the maxilla and mandible in relation to the skull base, the postoperative SNA showed no change when compared to the preoperative SNA. However, as for SNB and ANB, all cases showed improvement postoperatively. There were no statistically significant changes in the position of the hyoid bone, tongue, and anteroposterior width diameter of the mid-pharynx over time, preoperatively, 2 months postoperatively, and 6 months postoperatively (Figure 2).

### 3.2. VF

The mastication time tended to increase over successive periods. In contrast, the oral and pharyngeal transfer times tended to decrease (Figure 3).

### 3.3. MWST

In all patients, Score 5 results were obtained during the study period.

### 3.4. FT

Similar to the MWST, a Score of 5 was obtained for all patients during the study period.

### 3.5. RSST

In all patients, at least three salivary swallows were achieved within 30 s during the preoperative and postoperative periods (Figure 4).

## 4. Discussion

SSRO with fixation has been reported to provide good postoperative skeletal stability [15] and swallowing function 6 months postoperatively [9]. However, postoperative stability has already been reported for SSRO without rigid fixation [7]; nevertheless, postoperative eating and swallowing functions have not been clarified. In the present study, the results obtained with MWST, FT, and RSST showed that swallowing function was maintained postoperatively. The MWST, FT, and RSST are reproducible and simple indicators for swallowing [16] and are used to evaluate swallowing function in head and neck cancer patients [10]. On the other hand, it is rarely used in patients with jaw deformities. In this report, patients were young, and the fact that no surgery was performed on the hyoid bone or larynx that would impair swallowing function may have contributed to the good clinical outcome.

The position of the hyoid bone and tongue on cephalometric analysis and the anteroposterior width of the pharyngeal region did not change between the preoperative and postoperative periods. This indicates that the anatomical landmarks in cephalometric photographs may be associated with swallowing function. Furthermore, there was no difference in the position of the hyoid bone or width of the oropharynx two months postoperatively when compared to preoperatively. This could be due to the recovery of eating and swallowing functions and harmonization of the oropharyngeal region after surgery.

In VF, mastication time increased after surgery. The clinical significance of this result is that the improvement in occlusal condition enabled smooth mastication. In contrast, the oral transfer time and pharyngeal transfer time decreased after surgery. This may be because the patients were able to masticate more smoothly, resulting in better bolus formation. In particular, the statistically significant difference in oral transfer time between preoperatively and 6 months postoperatively suggests that as the occlusal condition stabilized, tongue function improved and the transition from mastication to swallowing became smoother. In terms of pharyngeal transfer time, little difference was found between the preoperative and postoperative time points, suggesting that the function of the pharyngeal region was maintained after surgery, similar to previous reports [9].

The position of the proximal segment after SSRO remains controversial and has not been strictly defined. Therefore, there is a risk of postoperative disorders of the temporomandibular joint when proper fixation is not performed [2]. To overcome this problem, long-term postoperative skeletal stability using a novel treatment without plates and screws was confirmed, and postoperative skeletal relapse and the development were suppressed [7]. The clear differences between this technique and IVRO are that this method does not require 4–6 weeks of intermaxillary fixation, and IVRO requires more time for bony healing owing to the small bony contact between the proximal and distal segments, therefore it is unclear whether the proximal segment is relocated to a physiologically appropriate functional position because the proximal segments are buccally elevated postoperatively by distal segments [6]. The significance of loose fixation is that the proximal segment can return to its physiologically natural position postoperatively. We evaluated the postoperative eating and swallowing functions in SSRO without fixation and obtained generally favorable results. In this study, stabilization of the occlusal condition was prioritized, and no evaluation of eating and swallowing functions was performed immediately after surgery. However, physiological recovery of eating and swallowing functions at two months postoperatively was achieved, which is consistent with a previous study wherein fixation was performed [9]. Mandibular setback without fixation did not interfere with postoperative swallowing function. The reason was thought to be that the pterygomassetieric sling, sphenomandibular ligament, stylomandibular ligament, and lateral pterygoid muscle attached to the proximal segment and the suprahyoid muscles including mylohyoid muscle, stylohyoid muscle, diagastric muscle, and geniohyoid muscle attached to the distal segment were not overstressed.

This study had some limitations. It has been reported that SSRO with loose fixation is an effective method, but it is not clear how to distinguish it from conventional SSRO and IVRO. In the future, through joint research with other institutions, it is necessary to verify which cases are suitable for SSRO with loose or no fixation. In addition, the number of patients was small; therefore, a larger number of cases will need to be studied in the future. The increased number of cases will allow for multifaceted evaluation, which will make it possible to determine what factors are independently involved in postoperative swallowing function.

Furthermore, it has not yet been clarified how the swallowing function of SSRO patients changes with age, and long-term follow-up is necessary to determine whether physiological function can be maintained.

## 5. Conclusions

In this study, good clinical outcomes were obtained in the evaluation of preoperative and postoperative eating and swallowing functions in SSRO with loose fixation. In the future, it will be necessary to increase the number of cases through joint research with other institutions and verify the indications for SSRO with loose fixation.

## Figures and Tables

**Figure 2 ijerph-20-01926-f002:**
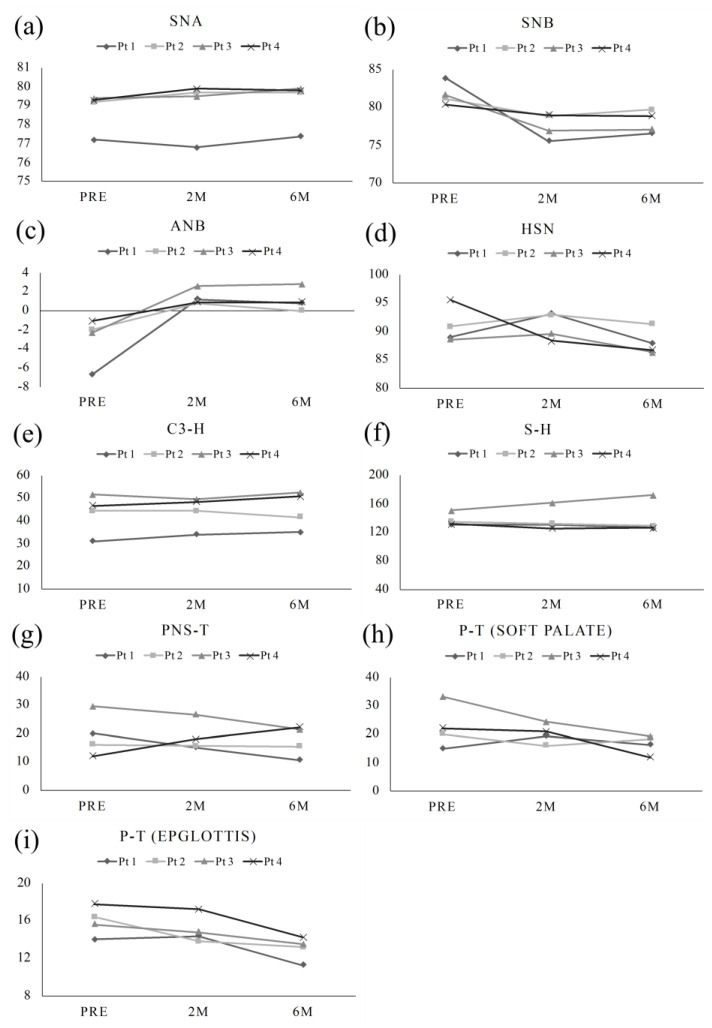
(**a**) SNA and (**b**) SNB show the position of the maxilla and mandible to the base of cranium over time, respectively. (**c**) ANB shows the transition of the anteroposterior position of the maxilla and mandible over time. (**d**) HSN, (**e**) S-H, (**f**) C3-H, and (**g**) PNS-T show the preoperative and postoperative hyoid and tongue position over time. And (**h**) P-T (soft palate) and (**i**) P-T (epiglottis) show the anteroposterior width of the pharyngeal region over time. As for maxilla and mandible, SNA did not change significantly preoperatively and postoperatively, while the SNB and ANB showed improvement postoperatively. The position of the hyoid bone and the anteroposterior diameter of the airway did not change significantly between preoperative and postoperative periods.

**Figure 3 ijerph-20-01926-f003:**
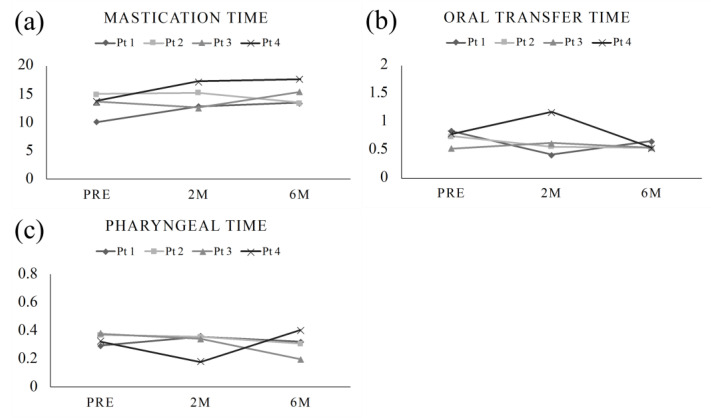
Transition of (**a**) mastication time, (**b**) oral transfer time, and (**c**) mastication time preoperatively, 2 months postoperatively and 6 months postoperatively. Mastication time showed a slight increase in the postoperative period, while oral transfer time and pharyngeal time showed no significant difference between the preoperative and postoperative periods.

**Figure 4 ijerph-20-01926-f004:**
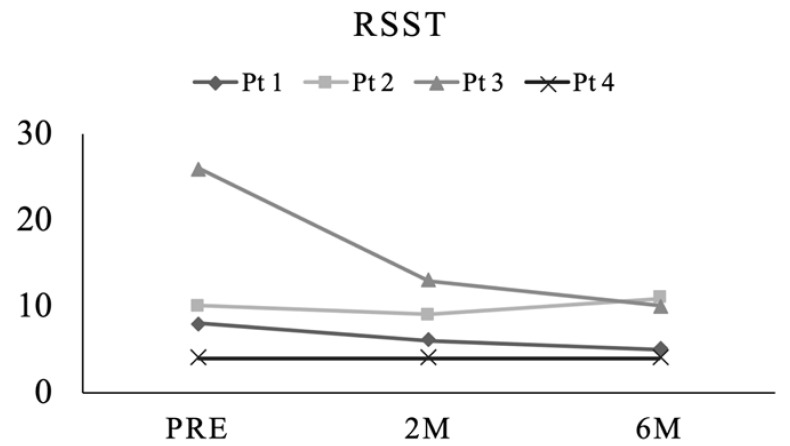
As for the RSST, more than three salivary swallows within 30 s were achieved throughout the preoperative and postoperative periods.

## Data Availability

Not applicable.

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
