# Peer review of "Swallowing Functions after Sagittal Split Ramus Osteotomy with Loose Fixation for Mandibular Prognathism: A Retrospective Case Series Research"

_ijerph, 2023, doi:10.3390/ijerph20031926_

Round 1

Reviewer 1 Report

This is an interesting follow up to previous studies examining the stability of fixation free orthognathic surgery

What it lacks is sufficient comparison with other studies - of either orthognathic surgery with fixation, or frankly any other maxillofacial surgery, to indicate that the methodology is appropriate and sound

I recommend you go back and do a further literature search focused on the swallowing studies you have used, and provide further discussion as to their suitability and interpretation of results

For the unfamiliar, a slightly more detailed explanation of physiological positioning strategy would not go amiss, and why the physiological positioning might affect swallowing.

There is some repetition with section 2.2 definitions mirroring what is in the figure legend - one is sufficient

Some tidying of English language required; eg repetition in line 162, line 64 comes to an abrupt end (I assume you mean none have had orthognathic 'surgery'.

Reviewer 2 Report

Thank you for the opportunity to review article titled “Swallowing functions after sagittal split ramus osteotomy with  loose fixation for mandibular prognathism: a retrospective case  series research”

The topic is interesting but the article needs extensive language and editorial correction.

First author should decide if this is loose or no- fixation SSRO and if it is the first one, explain it more in the introduction. Also in the introduction section authors should put more pressure on explaining the research methodology and justify the reasons for this research.  In the material and methods section the time frame of the research should be described first and inclusion/exclusion criteria should be added. With so few participants the single patient measures should be presented in table instead of mean values . There is no technical information about the equipment , programs nor the information who performed the radiological assessment .

The tables are hard to read and should be rewritten

Line 38-39 need references

Line 64 . None of the patients had a previous orthognathic - ?

Line 125-126  need more explanation

References need to be corrected , it would be better to use more updated literature.

In my opinion for a case report the number of self-citation is too high.

Round 2

Reviewer 1 Report

Much improved manuscript, thank you

My main criticism now would be minor, and just about the terminology used

Through this manuscript you have used the term; 'SSRO without rigid/semirigid fixation' which to me is a bit clunky - is this not the same technique your group previously described as 'physiological positioning strategy' ?  the latter is a much more catchy term, and I wonder if something based on that might not have been clearer and demonstrated more clearly that this is a follow on from that

The only other point would be the relatively small number of cases studies, which you have acknowledged at the end of the manuscript.  Maybe a small comment about why a larger number would help would strengthen this (it is rather easy to say more studies are needed with more cases - but in reality that is not always true (you only need a small number of people to jump out of a plane at height without a parachute to know it is not a good idea!) - I personally think it is better to add a comment about what might be different - is it body habitus / variations in MMPA, gender  / cervico mental angle etc? that might affect this that a larger number of cases might allow to be assessed as independant factors - keep it brief, but just show that you have thought where variations might come into your results

Reviewer 2 Report

The paper has improved a lot , authors have followed all reviewer sugestion.

Author Response

Thank you very much for your comments which will improve the quality of the article.